# Characterisation of Waterborne Psychrophilic *Massilia* Isolates with Violacein Production and Description of *Massilia antarctica* sp. nov.

**DOI:** 10.3390/microorganisms10040704

**Published:** 2022-03-24

**Authors:** Ivo Sedláček, Pavla Holochová, Hans-Jürgen Busse, Vendula Koublová, Stanislava Králová, Pavel Švec, Roman Sobotka, Eva Staňková, Jan Pilný, Ondrej Šedo, Jana Smolíková, Karel Sedlář

**Affiliations:** 1Department of Experimental Biology, Czech Collection of Microorganisms, Faculty of Science, Masaryk University, Kamenice 5, 625 00 Brno, Czech Republic; pavlah@sci.muni.cz (P.H.); 436839@mail.muni.cz (V.K.); kralova.s@sci.muni.cz (S.K.); mpavel@sci.muni.cz (P.Š.); evickakroupova@seznam.cz (E.S.); 2Institut für Mikrobiologie, Veterinärmedizinische Universität Wien, Veterinärplatz 1, A-1210 Wien, Austria; hans-juergen.busse@web.de; 3Centrum Algatech, Institute of Microbiology, Czech Academy of Sciences, Opatovický mlýn, 379 01 Třeboň, Czech Republic; sobotka@alga.cz (R.S.); pilny@alga.cz (J.P.); 4Central European Institute of Technology, Masaryk University, Kamenice 5, 625 00 Brno, Czech Republic; sedo@post.cz; 5Department of Physical Geography and Geoecology, Faculty of Science, Charles University, Albertov 6, 128 00 Praha, Czech Republic; janca.smolikova@gmail.com; 6Department of Biomedical Engineering, Faculty of Electrical Engineering and Communication, Brno University of Technology, Technická 12, 616 00 Brno, Czech Republic; sedlar@vut.cz; 7Institute of Bioinformatics, Department of Informatics, Ludwig-Maximilians-Universität München, Amalienstraße 17, 803 33 Munich, Germany

**Keywords:** *Massilia*, violacein, psychrophilic, description, whole-genome sequencing, Antarctica

## Abstract

A group of seven bacterial strains producing blue-purple pigmented colonies on R2A agar was isolated from freshwater samples collected in a deglaciated part of James Ross Island and Eagle Island, Antarctica, from 2017–2019. The isolates were psychrophilic, oligotrophic, resistant to chloramphenicol, and exhibited strong hydrolytic activities. To clarify the taxonomic position of these isolates, a polyphasic taxonomic approach was applied based on sequencing of the 16S rRNA, *gyr*B and *lep*A genes, whole-genome sequencing, rep-PCR, MALDI-TOF MS, chemotaxonomy analyses and biotyping. Phylogenetic analysis of the 16S rRNA gene sequences revealed that the entire group are representatives of the genus *Massilia*. The closest relatives of the reference strain P8398^T^ were *Massilia atriviolacea*, *Massilia violaceinigra*, *Massilia rubra*, *Massilia mucilaginosa*, *Massilia aquatica*, *Massilia frigida*, *Massilia glaciei* and *Massilia eurypsychrophila* with a pairwise similarity of 98.6–100% in the 16S rRNA. The subsequent *gyr*B and *lep*A sequencing results showed the novelty of the analysed group, and the average nucleotide identity and digital DNA–DNA hybridisation values clearly proved that P8398^T^ represents a distinct *Massilia* species. After all these results, we nominate a new species with the proposed name *Massilia antarctica* sp. nov. The type strain is P8398^T^ (= CCM 8941^T^ = LMG 32108^T^).

## 1. Introduction

The exploration of extreme habitats on our planet has become a challenge for the study of microbial diversity. Antarctica provides a unique opportunity to study the distribution of prokaryotic species in an environment where there are low temperatures and extreme seasonal fluctuations in light supply [1]. In Antarctic soil samples, some cultivable heterotrophic isolates belong to the genus *Massilia* [2,3]. The genus *Massilia* was established in 1998 by La Scola at al. [4], and includes bacteria of the family *Oxalobacteraceae*, class *Betaproteobacteria* [5]. The genus description was amended by Kämpfer et al. [6] and Singh et al. [7]. The cells of the members of the genus *Massilia* are non-spore-forming, predominantly motile rods, strictly aerobic, catalase-positive and show broad enzyme activity, implying that they could play a crucial role in carbon cycling in cold environments [3]. The oxidase reaction and acid production from carbohydrates can be variable; ubiquinone Q-8 is the predominant isoprenoid quinone [6].

Currently, the genus *Massilia* includes fifty-seven species with valid, published names [8] (http://lpsn.dsmz.de/genus/massilia; accessed on 1 January 2022), and these species are predominantly associated with various soil habitats [9,10]. Several species have been isolated from various environmental habitats, such as desert soil [11], rock surfaces [12], wolfram mine tailings [13], ice cores [14,15], glacial ice [16], drinking water [17], water [18], air samples [19] as well as from human clinical material [4,20]. *Massilia* spp. from Antarctica are reported in studies focusing on soils [3,21,22] or the aquatic environment [2,23]. Cold-adapted bacteria are able to produce a variety of pigments as a protective strategy against environmental stresses such as low temperature, oxidative stress, and ultraviolet radiation [24]. A positive effect of pigmentation in heterotrophic bacteria isolated from the Antarctic environment is that it increases resistance to environmental stressors [25]. *Massilia* spp. can produce different pigments, including yellow [17,26,27], pinkish-red [23] or blue-violet violacein [10,28]. The modelled results confirmed a positive effect of pigmentation on the survivability of heterotrophs [25]. In the genus *Massilia*, three species have been reported to produce violacein, namely *Massilia atriviolacea*, *Massilia flava* and *Massilia violaceinigra* [10,27,28], as a secondary metabolite. Violacein is a promising compound with antibacterial, antifungal, antiprotozoal and anticancer activity [29].

In our long-term studies on water-associated psychrophilic heterotrophs, a group of purple-pigmented strains was found from the environment, and a new species, *Rugamonas violacea,* was recently described [30] as part of the above-mentioned group. In this study, we describe a representative of a new species of the genus *Massilia*. All seven isolates were obtained from freshwater samples from James Ross Island and Eagle Island in Antarctica, and they exhibit the production of the pigment violacein, which makes them a potential source of natural pigment production. The studied group represents a new species with the proposed name *Massilia antarctica* sp. nov.

## 2. Materials & Methods

### 2.1. Isolates and Reference Strains

Freshwater specimens were analysed as previously described [30], and seven obtained pure purple pigmented strains were maintained at −70 °C until analysis. Details of the date of sampling, locality and source of isolates and coordinates of GPS are given in Table 1 and Appendix A. The reference cultures of the phylogenetically related *Massilia* spp. were obtained from the Czech Collection of Microorganisms: *M. atriviolacea* CCM 8999^T^, *M. violaceinigra* CCM 8877^T^, *M. eurypsychrophila* CCM 8735^T^, *M. glaciei* CCM 8861^T^, *M. rubra* CCM 8692^T^, *M. mucilaginosa* CCM 8733^T^, *M. aquatica* CCM 8693^T^ and *M. frigida* CCM 8695^T^. All the above types were used for the following comparison.

### 2.2. DNA Extraction, 16S rRNA, gyrB and lepA Genes Sequencing, Phylogenetic Analyses

DNA was extracted according to Kýrová et al. [31]. Amplification and sequence analysis of the partial 16S rRNA gene and two partial housekeeping genes encoding DNA gyrase subunit B (*gyr*B) and GTP-binding protein (*lep*A) were performed as previously described [19,23]. Sequencing was performed at the Eurofins Genomics sequencing facility (Ebersberg, Germany). The genomic techniques used were in accordance with the proposed minimum standards for prokaryote taxonomy [32].

Pairwise sequence alignment and the calculation of similarity scores [33] were used for the first classification of isolates at the genus level. The 16S rRNA gene sequence gained from the PCR product was aligned with the sequences extracted from the WGS data using the ContEst16S tool provided by EzBioCloud [34]. Phylogenetic analyses were performed using the software MEGA 11 [35]. Genetic distances were corrected using the two-parameter model of Kimura [36], and evolutionary history was inferred using the neighbour-joining method (NJ) and the maximum-likelihood method (ML). The GenBank/EMBL/DDBJ accession numbers for the analysed housekeeping gene sequences of the *Massilia* spp. cultures studied are given in Appendix A.

### 2.3. Genome Sequencing and Calculating Overall Genome Relatedness Index

The whole-genome sequencing of strain P8398^T^ and genome assembly with subsequent annotation were performed as previously mentioned [37,38]. Whole-genome sequencing of P8910 was performed by the Eurofins Genomics service facility. The average nucleotide identity (ANI) between the whole-genome sequences of strains P8398^T^ and P8910 and those of the phylogenetically closest *Massilia* spp. were calculated using OrthoANIu [39]. The genome-to-genome distance calculator (GGDC v2.1) [40] was used to calculate the Digital DNA-DNA hybridisation (dDDH) values, taking into account recommended formula 2.

### 2.4. Bioinformatics Analyses and Whole-Genome Phylogeny

Annotation of the genome, prediction of operons and functional annotation of protein-coding genes were performed as previously described [41,42,43]. PHASTER was used to search for prophage DNA [44]. The CRISPRDetect tool [45] was used to screen for the presence of Clustered Regularly Interspaced Short Palindromic Repeats (CRISPR) loci. Restriction modification systems were mapped using the REBASE database [46]. Metabolic pathways were mapped by searching for orthologues in KEGG [47] using BlastKOALA [48], while annotation was completed manually using BLAST searches [49]. The resistome was predicted with the Resistance Gene Identifier (RGI) 5.1.1 using the Comprehensive Antibiotic Resistance Database (CARD) 3.1.1 [50].

### 2.5. MALDI Analysis

Protein fingerprinting was performed by Matrix-assisted laser-desorption/ionisation time-of-flight mass spectrometry (MALDI-TOF MS) with an UltrafleXtreme instrument (Bruker Daltonics, Billerica, MA, USA) according to Freiwald and Sauer [51].

### 2.6. Chemotaxonomic Characterization

Polyamines were extracted as described by Busse and Auling [52] and analysed by HPLC using the data described by Busse et al. [53]. Polar lipids and quinones were extracted according to the integrated procedure of Tindall [54,55] and Altenburger et al. [56]. For the HPLC analyses, the apparatus described by Stolz et al. [57] was used. All compared strains were cultured for fatty acid methyl ester (FAME) analysis under identical conditions: Growth on R2A agar (Oxoid), at 20 °C ± 2 °C for three days. Once the cultures reached the late exponential growth stage according to the four-quadrant streak method, the biomass was harvested, and the FAMEs were extracted as described by Sasser [58]. For the subsequent analysis, an Agilent 7890B gas chromatograph with the Sherlock MIDI Identification System (MIDI Sherlock version 6.2, MIDI database RTSBA 6.21, Newark, USA) was used.

### 2.7. Repetitive Sequence-Based PCR Fingerprinting (rep-PCR)

Genomic relatedness between the isolated strains was investigated by rep-PCR fingerprinting analysis with the (GTG)_5_ primer according to Švec et al. [59]. The software BioNumerics version 7.6 (Applied Maths, Sint-Martens-Latem, Belgium) was used for numerical analysis and dendrogram construction.

### 2.8. Phenotypic Characteristics, Electron Microscopy and Violacein Determination

The Gram-staining results of the strains studied were read by light microscopy of stained cells from R2A agar after 72 h of cultivation at 20 °C and validated by the KOH test [60]. The cellular morphology of type strain P8398^T^ was also determined by transmission electron microscopy. The purple pigment violacein was detected as previously described [30]. Growth on R2A agar plates at different temperatures, different pH values or NaCl concentrations was determined as mentioned before [61,62]. Basic phenotyping at 20 °C was performed as previously published [61,63,64,65,66]. Additional biotyping with an API ZYM kit (bioMérieux, Marcy-l’Étoile, France) and a GEN III MicroPlate^TM^ kit (Biolog, Hayward, CA, USA) was performed [30] and enabled a comprehensive characterisation of the isolates. Differences in antibiotic resistance patterns were tested using twelve antibiotic discs [30], and EUCAST/CLSI standards were followed when reading the inhibition zone diameters [67,68]. All tests in this study were performed with two replicates.

## 3. Results & Discussion

### 3.1. 16S rRNA, gyrB and lepA Gene Sequencing and Phylogenetic Relationship

The group of seven isolated strains studied (Table 1) was first classified by sequencing the 16S rRNA gene. The similarities between the isolates examined ranged from 98.76% to 100% and assigned the isolates to the *Massilia* group. The most closely related species based on pairwise sequence alignment were *M. frigida* (99.9–100% similarity), *M. rubra* (99.8–99.9%), *M. violaceinigra* (99.7–99.9%), *M. mucilaginosa* (99.5–99.7%), *M. aquatica* (99.7–99.9%), *M. atriviolacea* (99.4–99.5%), *M. eurypsychrophila* (98.6–98.8%) and *M. glaciei* (98.8–99%) (Appendix A). Phylogenetic reconstruction based on the 16S rRNA gene sequence, carried out using the ML and NJ methods, divided all the studied isolates into a clade containing the species *M. frigida*, *M. rubra* and *M. violaceinigra*, with 81% bootstrap support (Appendix A). The results obtained prove that the isolates studied belong to the genus *Massilia*. Nevertheless, the 16S rRNA gene sequence similarities among the isolates and the *Massilia* spp. types were above the threshold of 98.7% and did not permit species discrimination. Thus, the partial *gyr*B and *lep*A gene sequences were additionally performed on all isolates and types. The thresholds for *Massilia* species differentiation were previously calculated to be 98% similarity for the *gyr*B gene and 99% similarity for the *lep*A gene [19]. In our study, the isolates made a distinct clade separating the new isolates from the known *Massilia* species (Figure 1). However, the sequences analysed had similarity values below the published threshold for the *gyr*B and *lep*A gene sequences, namely 100–97.2% for the *gyr*B gene and 100–97.8% for the *lep*A gene (Table 2). The sequencing similarity data indicate that the isolates belong to a new species with the representative P8398^T^ (CCM 8941^T^) as the type strain. Therefore, the ANI and dDDH values between P8398^T^ and P8910 were calculated based on whole-genome sequencing (Appendix A) to confirm that both isolates belong to a single species. These data were well above the threshold (Table 2) and confirmed the assignment of strains P8398^T^ and P8910 to a single taxon.

The ANI values were resolved and the intergenomic distances between the genome sequences of P8398^T^ and the representative types of *Massilia* spp. were analysed. The ANI values among P8398^T^ and *M. atriviolacea* CCM 8999^T^, *M. violaceinigra* CCM 8877^T^, *M. rubra* CCM 8692^T^, *M. mucilaginosa* CCM 8733^T^, *M. aquatica* CCM 8693^T^, *M. frigida* CCM 8695^T^, *M. glaciei* CCM 8861^T^ and *M. eurypsychrophila* CCM 8735^T^ ranged from 79.5% to 92.1%, and the dDDH values between the above strains ranged from 23.1% to 46.4, respectively (Table 2). The determined ANI and dDDH data between the studied strain P8398^T^ and *Massilia* spp. type strains differed from the threshold values of 95–96% (ANI) and 70% (dDDH) for bacterial species discrimination [32,39,40]. The obtained results clearly confirmed that strain P8398^T^ represents a distinct *Massilia* species.

### 3.2. Genome-Based Phylogeny and Basic Genome Characterization

The complete chromosomal sequence of strain P8398^T^ has been deposited at GenBank/EMBL/DDBJ under accession number CP065053. The version characterised in this study is CP065053.1. The complete genome has a size of 7.43 Mbp. The annotation predicted a total of 6509 genes. The majority of them (6392) were predicted to be protein-coding genes (CDSs), including 127 pseudogenes, 117 corresponded to RNA genes, including 22 genes for rRNAs (eight 5S, seven 16S and seven 23S rRNA), 91 genes for tRNAs and 4 genes for ncRNAs. The genomic G + C content was 63.8 mol%. Functional annotation of the CDSs in the form of division into clusters of orthologous genes (COG) assigned a specific function to the majority of CDSs (3956 genes, almost 62% of all CDSs), while the remaining genes belonged to the group without assignment or to the category (S) “Unknown Function”, representing 1337 and 1099 genes, respectively, suggesting that future research with the culture might reveal new, previously unknown traits (Appendix A). As for the next functional groups, most of the genes were related to groups responsible for housekeeping functions, such as (K) “Transcription”, (E) “Amino Acid transport and metabolism” and (M) “Cell wall/membrane/envelope biogenesis”, all containing at least 5% of the total number of genes. This threshold was only reached by another functional group (T) “Signal transduction mechanisms”, indicating the presence of a sophisticated system involved in responses to various intra- and extracellular stimuli. For this task, the cells can use 244 genes belonging to the (N) “Cell motility” group. Their further investigation revealed 86 KEGG orthologues belonging to the KEGG BRITE group ko02035 “Bacterial Motility Proteins”, which contains both flagellar and pilus systems.

Although no gene was associated with the class (X) “Mobilome: prophages, transposons”, two prophage sequences were identified in the P8398^T^ genome. While the first phage was marked as questionable, the second was intact (Appendix A). Nevertheless, 21 of its 54 genes coded hypothetical proteins, suggesting that the phage is relatively inactive or that it is a new phage that has not been reported before. Neither a CRISPR arrangement nor coupled *cas* or *cas*-like genes to create an adaptive immune system have been predicted in the genome [69]. However, this does not mean that P8398^T^ is completely devoid of systems that protect it from foreign DNA, as four restriction–modification (R-M) systems have been found in its genome whose function is, among other things, to defend against genome invasion [70] (Appendix A). While two type II systems are probably inactive, as predicted methylases lacked coupled restriction endonucleases, the remaining type I and type IV systems could be active. The type IV systems are formed by only one restriction enzyme identified in the P8398^T^ genome. Although the type I systems are more complex, this R-M system also appeared to be complete, containing one restriction, one modification and two specificity enzymes.

In the search for genes that cause antibiotic resistance, there was no perfect hit among the known genes. However, two copies of a putative *cat*B2 gene (locus tags IV454_00155 and IV454_32525) coding for chloramphenicol acetyltransferases were found among the loose hits, both with 38.1% sequence identity to the best hit. Since experimental work has demonstrated the strain’s resistance to chloramphenicol, it can be assumed that at least one of these genes is active in the genome of P8398^T^.

The psychrophilic nature of P8398^T^ and its ability to adapt to a hostile cold environment with freezing, thawing and strong UV irradiation is supported by a set of genes specialised in cold adaptation and several more versatile genes involved in the response to extreme conditions. In the first group, two copies of the *csp*A genes (IV454_17145 and IV454_22840) were found to encode cold shock proteins that play an important role in the immediate response to a decrease in temperature. We also identified some putative genes coding for anti-freeze proteins (IV454_32165, IV454_29925), which are likely to be involved in the synthesis or accumulation of cryoprotective substances. The main function of anti-freeze proteins is to reduce the freezing point of cell fluids in microorganisms to avoid freezing and to influence the shape and size of ice crystals [71,72]. Another important role is played by the genes for the composition of membrane fatty acids (unsaturated, polyunsaturated, methyl-branched, cis-form), which influence the fluidity of the membrane and the transition from the liquid crystalline to the gel phase [73]. The genome of P8398^T^ contained 16 genes coding for proteins of the OmpA (membrane transport) family. However, it is not possible to predict which of these genes are directly related to cold adaptation. Finally, we also found genes responsible for the synthesis of polyhydroxyalkanoates (PHAs), *pha*C (IV454_22210) and *pha*R (IV454_22215). PHAs play an important role in the stress resistance of bacteria, and also have the potential for industrial use, but are more commonly studied in other extremophiles, halophiles and thermophiles [74].

One of the most striking manifestations of the phenotype of P8398^T^ was the production of the purple pigment violacein. The genome of P8398^T^ contains all the genes necessary for its production. In addition, the genes *vio*A (IV454_26305), *vio*B (IV454_26310), *vio*C (IV454_26315), *vio*D (IV454_26320) and *vio*E (IV454_26325) were found in a single predicted *vio*ABCDE operon. Additional genes coding for transcription factors for violacein biosynthesis were also found. The genes *jqs*S/*cqs*S (IV454_01070), coding for the sensor kinase, and *jqs*R/*cqs*S (IV454_01075), coding for the response regulator, formed one operon, while *jqs*A/*cqs*A (IV454_01065), coding for a quorum-sensing autoinducer synthase, was a separate gene. Therefore, we are certain that the expression of the violacein biosynthetic operon in P8398^T^ depends on quorum sensing, as recently reported for *Rugamonas violacea*, another psychrophilic species from Antarctica [30].

The strain P8398^T^ exhibited strong gelatinase activity corresponding to the presence of the *col*A (IV454_08230) gene, which codes for a microbial collagenase. The hydrolytic activity of the strain is even broader. DNase activity is caused by a series of genes coding for exodeoxyribonucleases that produce 5′-phosphomonoesters: *sbc*B (IV454_15565) coding for exodeoxyribonuclease I, three copies of *xth*A (IV454_12835, IV454_17225, IV454_27120) coding for exodeoxyribonuclease III, *rec*D (IV454_27425), *rec*B (IV454_27430), *rec*C (IV454_27435) coding for alpha, beta and gamma subunits of exodeoxyribonuclease V located in a single operon on the antisense strand and *xse*A (IV454_16715) and *xse*B (IV454_13465) coding for large and small subunits of exodeoxyribonuclease VII. On the other hand, no exodeoxyribonucleases producing 3′-phosphomonoesters were found in the genome. Lipase activity is caused by three copies of putative *lip* genes (IV454_06595, IV454_31055 and IV454_31865) coding for triacylglycerol lipases, *pld*A (IV454_06250) coding for phospholipase A and *tes*A (IV454_24625) coding for acyl-CoA thioesterase I. The ability to hydrolyse lecithin corresponded to the presence of three copies of the gene *plc* (IV454_12025, IV454_20950 and IV454_24185), which codes for phospholipase C. The strain also exhibited glycolic activity. The genome contained, among others, the gene *amy*A (IV454_01825, IV454_06345), coding for alpha-amylase, in two copies and the genes *bgl*X (IV454_05025, IV454_11505, IV454_17410) and *bgl*B (IV454_13665, IV454_19850, IV454_27750), coding for beta-glucosidase, in three copies each. In contrast, we could not find a gene responsible for the tyrosinase activity observed at the phenotype level.

The whole-genome sequence of strain P8910 (= CCM 9189) has been deposited with DDBJ/ENA/GenBank under accession number JAJTSQ000000000, and the version described in our study is JAJTSQ00000000.1. The resulting draft genome was 7.4 Mb in size with 190 contigs (N50 = 115,275; L50 = 20). Annotation revealed 6687 genes with 6474 protein-coding genes (CDS), 87 RNA genes including 8 genes for rRNAs (16S and 23S rRNA) and 79 genes for tRNAs. The genomic G + C content was 63.5 mol%, which is fully consistent with the percentages known for other *Massilia* spp. [6,7].

### 3.3. MALDI-TOF MS

The MALDI-TOF MS protein fingerprints of the seven examined isolates showed no significant similarity to any of the entries in the Bruker Biotyper database (version 10.0, 9607 entries). After expanding this database with protein signals from eight type strains of *Massilia* spp. including strain P8398^T^, all remaining six isolates could be correctly assigned to the corresponding species, resulting in log (scores) greater than 2.0. The mutual similarity between the examined isolates is confirmed by a dendrogram showing their cluster separated from the rest of *Massilia* spp. (Figure 2). The slightly divergent branch of P8910 was assigned to a common cluster according to the whole-genome sequencing results from ANI and the dDDH data. Visual inspection of the mass spectra revealed that all seven isolated strains shared 24 signals in the *m*/*z* range from 2500 to 10,000, while only two of these peaks were specific to the seven isolates (*m*/*z* = 3227 and 6453, which most likely corresponds to different ionisation forms of the same compound), allowing their differentiation from the rest of the *Massilia* spp. tested.

### 3.4. Chemotaxonomic Characterization

The polar lipid profile (Appendix A) of culture P8398^T^ consisted of the main lipids diphosphatidylglycerol, phosphatidylglycerol and phosphatidylethanolamine. In addition, small quantities of an ornithine lipid (OL) and several unidentified lipids were detected, including an aminolipid (AL1) and nine lipids with no detectable functional group (L1-L9), as well as a yellow pigment spot. The polar lipid profile of strain P8398^T^ did not show the presence of two unidentified phospholipids reported to be present in *M. violaceinigra* and *M. atriviolacea* [10,28]. The quinone system contained only ubiquinone Q-8, which is a feature of other *Massilia* species. The polyamine pattern consisted of 4.8 µmol (g dry weight)^−1^ putrescine, 0.5 µmol (g dry weight)^−1^ 2-hydroxyputrescine, 0.3 µmol (g dry weight)^−1^ spermidine and 0.1 µmol (g dry weight)^−1^ cadaverine. A comparable polyamine pattern was described for *Massilia norwichensis* [19].

Analysis of the fatty acid methyl esters showed that the major fatty acids of all strains were uniformly iso-C_16:0_ and inseparably C_16:1_
*ω7c*/C_16:1_
*ω6c* (included in Summed Feature 3) via the Sherlock MIDI system, with an average abundance in the total fatty acid profiles (FA) of 24.8% and 55.4%, respectively (Appendix A). The presence of these major FAs corresponded to the FA profiles of the most closely related *Massilia* spp. All strains belonging to the newly described species could be distinguished from their closest phylogenetic neighbours by the presence of capric acid (C_10:0_). Apart from the presence of capric acid, all *Massilia* spp. analysed in this study had similar FA profiles with insignificant quantitative differences, confirming their relatedness.

### 3.5. Repetitive PCR-Based Fingerprinting

The cluster analysis of the rep-PCR fingerprint profiles of the strains studied is shown in Appendix A. The four strains P8323, P9735, P8910 and P11689 had visually distinct fingerprints that enabled their clear differentiation. The fingerprints of the remaining three strains P8398^T^, P9640 and P11691 were visually similar, but minor differences due to the absence or presence of certain bands also allowed them to be distinguished. Interestingly, these three strains came from different, distant localities. These results showed that the rep-PCR band patterns obtained enabled fine differentiation of the isolates at the subgroup level and proved that they are not clonally related.

### 3.6. Phenotypic Characteristics and Description of Novel Species

Description of *Massilia antarctica* sp. nov. (*M. antarctica* etymology: *ant.arc´ti.ca.* L. fem. adj. *antarctica* southern, pertaining to Antarctica, the geographical origin of the isolated strains).

The cells are Gram-stain-negative rods, occurring separately or in clusters, motile with a long polar flagellum (Appendix A). The width of the cells ranged from 880 nm to 1.14 µm and the length from 1.68 µm to 2.93 µm. The colonies on the R2A medium are circular, with entire margins, convex, smooth, glistening with blue-purple endopigment violacein, 2 mm in diameter after 72 h of cultivation at 20 °C and moderately slimy (Appendix A). The results of the biochemical and physiological tests, which enabled the phenotypic distinction of the investigated strains from closely related species, are described in Table 3. The strains were resistant to chloramphenicol, and all exhibited strong hydrolytic activities. The detailed results of the phenotypic characterization of the new species *M. antarctica* sp. nov. are given in the Protologue species description (Box 1) and the variable results are given in Appendix A. The type strain is P8398^T^ (= CCM 8941^T^ = LMG 32108^T^).

Box 1Protologue species description—biochemical and physiological properties of *Massilia antarctica* sp. nov. All presented data are uniform for all seven isolates, the strain dependent data are shown in Appendix A.
**Phenotype characterisation**
Aerobic growth only occurs on R2A agar and PCA
agar. No growth was observed on TSA, BHI, Mac Conkey agar, Mueller–Hinton
agar or Nutrient agar at 20 °C. Growth is observed between 1 °C and 25 °C,
but not at −2 °C, nor at 30 °C. Cells grow well in the pH range 6.0–9.0, and pH
5.0 and pH 10.0 inhibits growth. Good growth on the R2A medium occurs without
NaCl, while the presence of 0.5% NaCl (*w*/*v*) inhibits growth. Acid production (aerobic) from glucose, fructose,
xylose and maltose occurred, while the fermentation of glucose in the OF test
medium is negative. Catalase, alkaline phosphatase, esterase lipase and
leucine arylamidase were positive by API ZYM. Gelatine, casein, esculin,
Tween 80, ONPG, starch and tyrosine hydrolysis positive. Lipase (C14), cystine arylamidase, α-chymotrypsin, α-galactosidase,
β-galactosidase, β-glucuronidase, N-acetyl-β-glucosaminidase, α-mannosidase
and α-fucosidase were negative by API ZYM. Oxidase, nitrite reduction,
fluorescein (King B medium), urease, lysine and ornithine decarboxylase,
arginine dihydrolase, Simmons citrate, malonate utilisation and acetamide
utilization were negative. All strains were negative for the utilization
(Biolog) of D-maltose, D-trehalose, gentiobiose, sucrose, turanose,
stachyose, D-raffinose, α-D-lactose, D-melibiose, β-methyl-D-glucoside,
D-salicin, α-D-glucose, D-mannose, N-acetyl-D-glucosamine, N-acetyl-β-D-mannosamine,
N-Acetyl-D-galactosamine, N-acetyl neuraminic acid, D-fructose, D, galactose,
3-methyl glucose, D-fucose, L-fucose, L-rhamnose, inosine, D-sorbitol,
D-mannitol, D-arabitol, myo-inositol, glycerol, D-aspartic acid, D-serine,
gelatine, glycyl-L-proline, L-alanine, L-arginine, L-aspartic acid,
L-glutamic acid, L-histidine, L-pyroglutamic acid, L-serine, pectin,
D-galacturonic acid, D-galactonic acid lactone, D-gluconic acid, D-glucuronic
acid, mucic acid, D-saccharic acid, p-hydroxy phenylacetic acid, methyl
pyruvate, D-lactic acid methyl ester, L-lactic acid, citric acid, α-keto
glutaric acid, D-malic acid, bromo-succinic acid, Tween 40, γ-amino-butyric
acid, α-hydroxy-butyric acid, propionic acid, acetic acid and formic acid as
carbon sources. They were resistant to chloramphenicol but
sensitive to ciprofloxacin, gentamicin, imipenem, kanamycin, cotrimoxazole,
piperacillin, streptomycin and tetracycline.

## Figures and Tables

**Figure 1 microorganisms-10-00704-f001:**
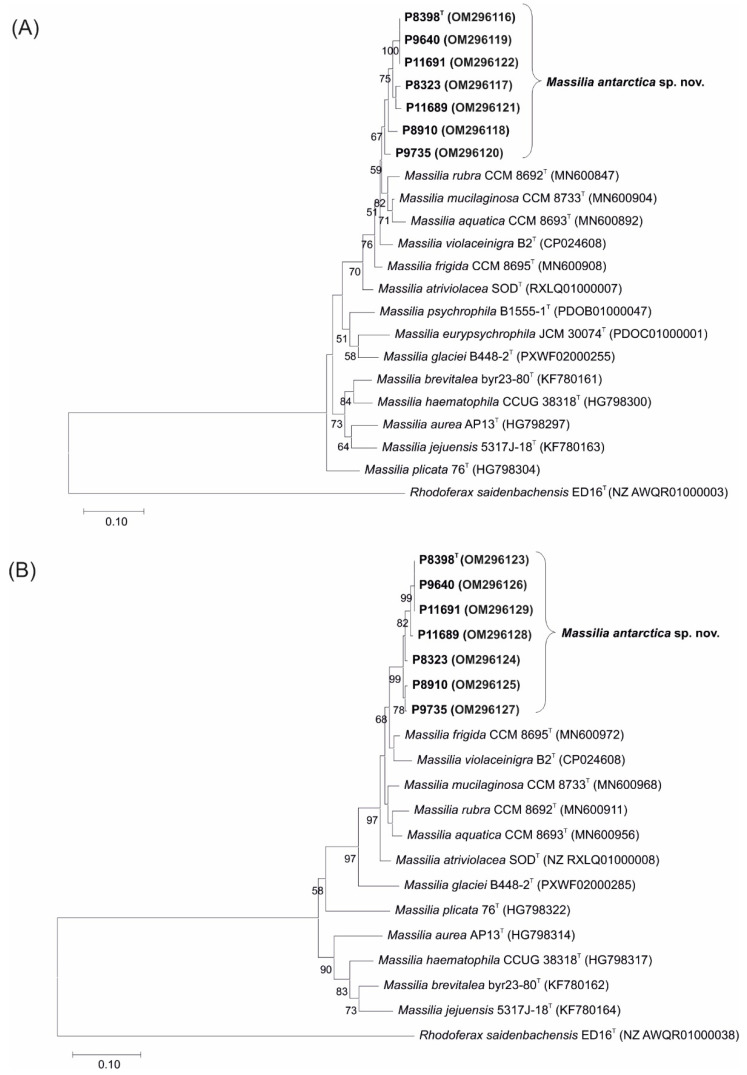
Phylogenetic maximum likelihood trees based on (**A**) *gyr*B and (**B**) *lep*A gene analyses showing phylogenetic positions of *M. antarctica* sp. nov. isolates and closely related *Massilia* spp. type strains. Bootstrap probability values (percentages of 500 tree replications) greater than 50%, in which the allied species are clustered, is shown next to the branches. The tree is drawn to scale, with the length of branches measured in the number of substitutions per site. *Rhodoferax saidenbachensis* D16^T^ sequences were used as an outgroup. GenBank accession numbers of the sequences are indicated in parentheses. Bar, 0.1 substitutions per nucleotide position.

**Figure 2 microorganisms-10-00704-f002:**
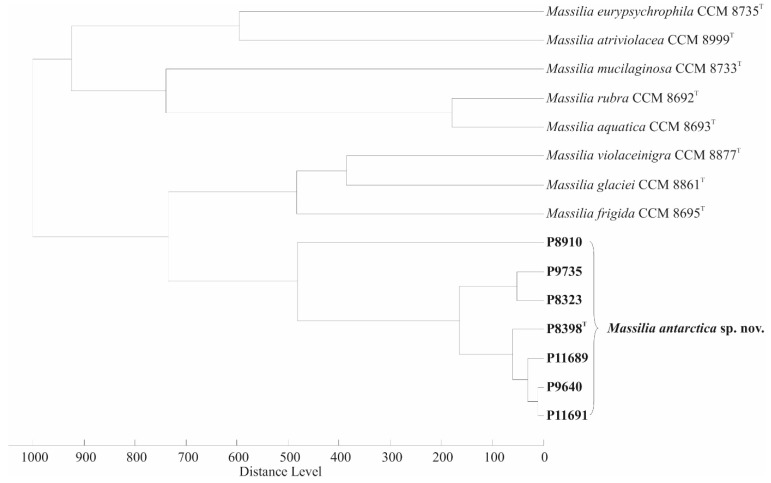
Dendrogram obtained by cluster analysis of MALDI-TOF mass spectra of *M. antarctica* sp. nov. isolates and the closest related *Massilia* spp. types, generated with Biotyper 3.0 software using Pearson’s product-moment coefficient as a measure of similarity and the unweighted pair group average linked method (UPGMA) as a grouping method. The distance is shown in relative units.

**Table 1 microorganisms-10-00704-t001:** Source of *Massilia antarctica* sp. nov. strains.

Strain Number	Year of Isolation	Locality (GPS)
P8323 (CCM 8944)	2017	Seals stream, nearby Monolith lake, James Ross Island (63°53′10″ S, 57°56′50″ W)
P8398^T^ (CCM 8941^T^)	2017	Glacial stream, Komárek valley, James Ross Island (63°49′51″ S, 57°49′48″ W)
P8910 (CCM 9189)	2017	Small stream nearby Whiskey glacier, James Ross Island (63°55′03″ S, 57°56′25″ W)
P9640	2018	Lakelet nearby sea coast, Eagle Island (63°37′29″ S, 57°25′52″ W)
P9735	2018	Lakelet below moraine Triangular, James Ross Island (63°51′42″ S, 57°51′07″ W)
P11689	2019	Small lake, approx. 200 m SW from Rožmberk lake, James Ross Island (63°49′17″ S, 57°50′28″ W)
P11691	2019	Small lake, approx. 200 m SW from Rožmberk lake, James Ross Island (63°49′17″ S, 57°50′28″ W)

**Table 2 microorganisms-10-00704-t002:** Sequence similarity values of analysed genes, average nucleotide identities and the dDDH values (%) of *Massilia antarctica* sp. nov. reference strains P8398^T^ (= CCM 8941^T^) and P8910 (= CCM 9189) to related *Massilia* spp. All data were taken from this study.

Species	Strain	Gene Sequence Similarities of P8398^T^			ANI and dDDH Values of P8398^T^		Gene Sequence Similarities of P8910			ANI and dDDH Values of P8910	
		16S rRNA	*gyr*B	*lep*A	ANI	dDDH	16S rRNA	*gyr*B	*lep*A	ANI	dDDH
*M. antarctica* sp. nov.	P8398^T^	100	100	100	100	100	99.9	97.0	97.8	97.0	73.9
*M. antarctica* sp. nov.	P8910	99.9	97.0	97.8	97.0	73.9	100	100	100	100	100
*M. aquatica*	CCM 8693^T^	99.7	95.8	95.0	87.6	33.7	99.8	96.0	95.0	87.7	33.7
*M. atriviolacea*	CCM 8999^T^	99.5	94.4	95.1	87.9	34.0	99.5	94.5	94.9	87.7	33.8
*M. eurypsychrophila*	CCM 8735^T^	98.8	89.1	88.2	79.3	23.1	98.7	89.1	87.4	79.2	23.0
*M. frigida*	CCM 8695^T^	100	96.0	96.4	92.1	46.4	99.9	96.4	96.4	92.2	46.3
*M. glaciei*	CCM 8861^T^	98.9	90.9	88.9	79.0	23.4	99.0	90.7	88.5	79.0	23.1
*M. mucilaginosa*	CCM 8733^T^	99.7	96.5	95.5	88.0	34.7	99.6	96.6	95.1	88.0	34.6
*M. rubra*	CCM 8692^T^	99.9	95.4	94.9	88.0	34.4	99.9	95.5	94.5	88.1	34.4
*M. violaceinigra*	CCM 8877^T^	99.9	95.4	95.5	91.0	42.4	99.9	95.7	95.3	91.0	42.1

**Table 3 microorganisms-10-00704-t003:** Distinguishing properties of *Massilia antarctica* sp. nov. with types of nearest phylogenetically related species. 1. *Massilia antarctica* sp. nov., 2. *M. rubra* CCM 8692^T^, 3. *M. mucilaginosa* CCM 8733^T^, 4. *M. aquatica* CCM 8693^T^, 5. *M. frigi*da CCM 8695^T^, 6. *M. violaceinigra* CCM 8877^T^, 7. *M. atriviolacea* CCM 8999^T^, 8. *M. glaciei* CCM 8861^T^, 9. *M. eurypsychrophila* CCM 8735^T^. +, positive; w, weak; -, negative; S, sensitive; R, resistant; all data were taken from this study.

Test	1 *	2	3	4	5	6	7	8	9
Violet pigment	+	-	-	-	-	+	+	-	-
Pink-red pigment	-	+	+	+	+	-	-	-	-
Oxidase	-	-	-	-	-	-	-	+	+
Growth at 5 °C on R2A	+	+	+	+	+	-	+	+	+
Growth in 0.5% NaCl	-	-	+	+	-	-	w	-	-
Acid from xylose	+	+	w	-	-	-	+	-	-
API ZYM: Lipase (C14)	-	-	-	-	-	+	w	-	+
Cystine arylamidase	-	-	-	-	-	+	-	-	+
BIOLOG GEN III: Arabitol	-	-	-	-	-	-	+	-	-
Pectin	-	-	-	-	-	-	+	-	-
Glucuronamide	-	+	+	w	+	+	-	-	+
Chloramphenicol (30 µg)	R	S	R	S	S	S	R	S	S
Hydrolysis of: Gelatine	+	+	+	+	+	+	+	+	-
Starch	+	-	+	-	+	+	+	+	-
ONPG	+	+	-	-	+	+	-	+	-
Tween 80	+	+	+	+	-	+	+	-	+
Tyrosine	+	+	+	+	+	+	+	-	-

* data are uniform for all isolates of *M. antarctica* sp. nov.

## Data Availability

The genomes of two isolates (P8398^T^ and P8910) were submitted to the GenBank database under accession numbers CP065053 and JAJTSQ000000000, respectively. The GenBank accession numbers (16S rRNA, *gyr*B and *lep*A genes) of *M. antarctica* sp. nov. strains are OM243916–OM 243922, OM296116-296122 and OM296123-OM296129, respectively.

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
