# Peer review of "Characterisation of Waterborne Psychrophilic Massilia Isolates with Violacein Production and Description of Massilia antarctica sp. nov."

_microorganisms, 2022, doi:10.3390/microorganisms10040704_

Round 1

Reviewer 1 Report

The article by Ivo Sedlacek et al. is devoted to the characterization of individual representatives of the psychrophilic aquatic bacterial species Massilia. The authors presented a qualitative work, including the molecular biological and biochemical characterization of one of the isolated strains.

The manuscript corresponds to the profile of the journal microbiology and can be accepted for publication. I have only two comments. Page 8, line 295. The authors state that strain P8398t exhibits strong gelatinase activity. On what basis do the authors draw a such conclusion? To make a decision about the magnitude of activity, the authors need to give its quantitative characteristics. There are no results in this manuscript that would allow us to draw a conclusion about the quantitative value of the activity.

Section 3.4. It seems to me that in this section it would be appropriate to give a link to Table S5.

Author Response

The revieweer had two comments:

  •  Page 8, line 295. The authors state that strain P8398t exhibits strong gelatinase activity.
    • The hydrolysis of gelatin was tested by a conventional plate test without quantitative data and the "strong gelatinase activity" was based on my long experience with this technique, i.e. a large, clear hydrolysis zone around the colony in a short time. But if you wish I can remove "strong".
  • Section 3.4. It seems to me that in this section it would be appropriate to give a link to Table S5. 
    • I do not understand this comment because in section 3.4. in the second paragraph is the link to Table S5. 

Reviewer 2 Report

Manuscript ID microorganisms-1631156 – review

The manuscript entitled “Characterisation of waterborne psychrophilic Massilia isolates with violacein production and description of Massilia antarctica sp. nov.” by Ivo Sedláček et al. describes properties of seven water-associated, purple pigmented psychrophilic, heterotrophs bacterial strains, from James Ross Island and Eagle Island in Antarctica. The authors identified the isolates as Massilia spp using classical culture methods, molecular identification and phenotypic fingerprinting using the Biolog GEN III test and MaldiTOF MS technique, as well as distinguished and characterized a new species of Massilia antarctica.

Substantially justified manuscript contains new content and research results. Editorial correct paper has all the necessary chapters and is easy to read. The title is appropriate to the content and the paper contains comprehensive research documentation. References justified and used properly.

This subject matter is important and interesting from the point of view of environmental microbiology, especially for understanding how those unique polar ecosystems work.

Below are some of my specific comments:

Perhaps it is worth including photos of colony growth and morphology on R2A medium in the additional materials, as well as describing the physical and chemical parameters of the water environment from which the described strains of bacteria were obtained.

Figure S1 is of very poor quality.

Figure S6 - Can you explain what is this oval vesicular structure at the base of the flagellum?

Author Response

The reviewer had four comments:

  • Perhaps it is worth including photos of colony growth and morphology on R2A medium
    • The photo of the morphology of the colonies was taken and included in the Supplementary material (Fig. S6b).
  • describing the physical and chemical parameters of the water environment
    • Unfortunately, no data are available to describe the physical and chamical parameters of the water environment.
  • Figure S1 is of very poor quality
    • Fig. S1 has been replaced with the figure of higher quality.
  • Figure S6 - Can you explain what is this oval vesicular structure at the base of the flagellum?
    • It is probably a buble attached to the surface of the cell (there is no cell structure clearly bounded by the membrane).